# Demographic variation in symptoms of depression and anxiety across 22 Global Flourishing Study countries

Matt Bradshaw [1] ✉, Koichiro Shiba[2], Sung Joon Jang [3,4], Blake Victor Kent [5,6], Rebecca Bonhag [3], Byron R. Johnson[3,4,7] & Tyler J. VanderWeele [7,8]

## Abstract

**Background** We know relatively little about how mental health varies across countries around the world or among demographic groups in diverse nations and cultures. **Methods** The current study addresses these issues by analyzing symptoms of depression and anxiety using data from the Global Flourishing Study (GFS), an international, nationally-representative survey of 202,898 individuals from 22 geographically, economically, and culturally diverse countries collected in 2022-2023. **Results** Here we show that proportions of individuals with substantial symptoms of depression range from 0.14 in Poland to 0.50 in the Philippines. These two countries report the lowest and highest substantial symptoms of anxiety as well (0.13 and 0.48, respectively). Lower-income, non-Western countries tend to have higher proportions of both outcomes compared with higher-income, predominantly Western nations. Symptoms of depression and anxiety also vary across age, gender, marital status, education, employment status, religious service attendance, and immigration status in one or more countries. The results of random effects meta-analyses show that several demographic factors are significant predictors of both outcome variables when the results for all 22 countries are pooled. **Conclusions** While being mindful of varying cultural contexts and possible translation and interpretive issues with the survey questions, the results suggest substantial variations in symptoms of both depression and anxiety across nations and key demographic groups. This work lays the foundation for future longitudinal GFS studies of mental health from a cross-national and global perspective.

## Plain language summary

This study examines how mental health varies in countries around the world and among demographic groups in diverse nations and cultures. Data from a nationally-representative survey of 202,898 individuals from 22 geographically, economically, and culturally diverse countries is analyzed. The data was collected in 2022-2023. The percentage of individuals reporting substantial symptoms of depression ranges from 14%-50% in different countries, while anxiety ranges from 13%-48%. Symptoms of both outcomes also vary across age, gender, marital status, education, employment status, religious service attendance, and immigration status in one or more countries. These findings highlight considerable variation across countries in mental health, as well as important demographic differences, which identify vulnerable populations that can be targeted with interventions.

Almost a billion people worldwide (about 13% of the population) suffer from mental illnesses, and this costs the global economy trillions of dollars each year[1]. According to the World Health Organization (WHO), median annual government spending on mental health per capita for countries that provided data in 2020 was $7.49, or 2.1% of the median for health overall[2]. Noting the meager and inadequate resources that governments commit to mental health, a recent global return on investment analysis concluded that treatment needs are not being met, and that mental illnesses affect health and well-being, family life, worker productivity, labor force participation, healthcare expenditures, and tax revenues[3].

Preventing mental illnesses before they develop is more cost-effective than treatment[4], so identifying risk and protective factors is crucial. We know that symptoms of depression, anxiety, and other indicators of mental health vary around the world[5–7], and that they are shaped by a variety of factors including: (a) demographic characteristics such as age[8], gender[9], and race/ethnicity[10]; (b) social relationships, integration, and support[11,12]; (c)

[1]Direct correspondence and reprint requests to: Matt Bradshaw, Institute for Global Human Flourishing and Institute for Studies of Religion, Baylor University, Waco, TX, USA. [2]Department of Epidemiology, Boston University, Boston, MA, USA. [3]Institute for Global Human Flourishing and Institute for Studies of Religion, Baylor University, Waco, TX, USA. [4]School of Public Policy, Pepperdine University, Malibu, CA, USA. [5]Department of Sociology & Anthropology, Westmont College, Santa Barbara, CA, USA. [6]Center on Genomics, Vulnerable Populations, and Health Disparities, Massachusetts General Hospital/Harvard Medical School, Boston, MA, USA. [7]Human Flourishing Program, Institute for Quantitative Social Science, Harvard University, Cambridge, MA, USA. [8]Department of Biostatistics and Department of Epidemiology, Harvard T.H. Chan School of Public Health, Boston, MA, USA. ✉e-mail: drmattbradshaw@gmail.com

socioeconomic status (SES)[13]; (d) cultural influences like religious practices and beliefs[14]; (e) stressful life events and chronic difficulties[15]; and (f) macrolevel contexts like political stability, economic security, and environmental concerns[1]. Our knowledge is limited in several ways, however. Many cross-national comparisons are simply reviews or meta-analyses of research on individual countries that use data and methods that may not be directly comparable[16–18]. With respect to actual cross-national studies, a number are confined to specific regions (e.g., Europe) or focus primarily on higher income nations[13,19,20]. Further, much of the truly global research is based on the World Mental Health (WMH) surveys, which were designed to facilitate comparisons by standardizing research designs across countries[21]. These studies have documented the prevalence and correlates of diagnosable mental disorders in multiple nations[7,22,23]. Despite many strengths (e.g., probability samples, validated instruments, standardized procedures)[21], these surveys have limitations. They focus primarily on diagnoses of mental disorders based on the WHO Composite International Diagnostic Interview (CIDI) and DSM-IV criteria[24], rather than broader symptoms that do not meet disorder-level thresholds[7,21–23]. While less severe, the latter are more common and may contribute to personal struggles and lower quality of life[25]. Several WMH samples are based on specific regions, states, or urban areas instead of entire populations (e.g., Argentina, Brazil, Japan, Mexico, Nigeria, and Spain)[21,26], which could limit generalizability. The surveys were conducted from 2001 to 2022[26], so changes in personal experiences or conditions over time could impact comparisons. In addition, several of the variables examined here (e.g., religious practices and identity, immigration status, race/ethnicity) have received very little attention in cross-national research on mental health.

The Global Flourishing Study (GFS) aims to address these limitations by providing an intended five waves of nationally-representative panel data on many aspects of mental health in 22 diverse nations around the world[27–29]. The first wave of annual data is publicly available, and the second is nearing completion. The survey includes countries that: (a) are geographically dispersed across North America, South America, Europe, Asia, Africa, and Oceania; (b) represent high, upper-middle, lower-middle, and low income nations; and (c) are culturally and religiously diverse, including nations with majorities of Christianity, Islam, Hinduism, and Judaism. Importantly, nine countries/territories that are not represented in the WMH surveys—Egypt, Hong Kong, India, Indonesia, Kenya, Sweden, Tanzania, Turkey, and the United Kingdom—are included in the GFS. Very little research has been conducted in several of these nations, so findings will expand our knowledge to new parts of the world. The current study has two objectives, which were revised based on research questions preregistered with the Center for Open Science (COS).

The first objective is to estimate the prevalence of two indicators of mental health—symptoms of depression and anxiety—in the 22 participating countries. These estimates will provide information on the proportion of individuals in each country who report relatively high levels of symptoms that may or may not meet the criteria for a diagnosis, but instead indicate a potential need for additional screening and assessment. It is likely that these estimates will be considerably higher than the diagnosable mental disorders reported in WMH studies[7,22,23,30] because they will include individuals who have, or may be at risk for developing, a depressive or anxiety disorder.

The second objective is to examine the associations between symptoms of depression and anxiety and a variety of demographic, socioeconomic, and religious characteristics that may function as risk or protective factors. Importantly, these may vary across nations. For example, age may have unique associations with mental health in different countries[22,26], but since there is a relative dearth of research based on nationally-representative samples in many nations, our knowledge remains limited. Likewise, research indicates that mental health problems tend to be more common among women compared with men[31], but a lack of quality data from numerous countries, particularly lower-income ones, makes it difficult to draw firm conclusions across cultures at this time. Family structure is another key correlate of mental health, but factors such as marital status may work in unique ways in nations with differing norms and values[22]. Indicators

of SES (e.g., education, income, employment) are also associated with mental health, but their effects may not be the same in economically, culturally, and geographically diverse countries in the global economy[5,19,20]. A growing number of studies have also linked religious involvement with mental health[14,32], but most research is based on higher-income, Western, and predominantly Christian nations, and scholars are just beginning to examine other countries around the world. Immigration status[33] and race/ethnicity[34–36] are important predictors of mental health as well, but we know little about how they function in different nations and cultures.

To date, numerous studies of risk/protective factors for mental health have been conducted in Germany[37], Japan[38,39], Sweden[40,41], the United Kingdom[42,43], and the United States[44,45]. A handful of publications have also focused on Australia[46,47], India[48,49], Mexico[50], South Africa[51], Hong Kong[52], Israel[53,54], and Spain[55], with a few in Brazil[56], Egypt[57], Indonesia[58], Nigeria[59], the Philippines[60], and Poland[61] as well. Virtually no research has been conducted in Argentina, Kenya, Tanzania, or Turkey. Importantly, many of these studies are based on individual countries and are not cross-national in nature, and most have examined a relatively small number of risk/protective factors. The current study contributes to this literature by examining nine key characteristics that may shape global patterns of mental health (age, gender, marital status, employment, education, religious attendance, immigration, religious tradition, and race/ethnicity) in all 22 countries combined and within individual nations to identify unique country-specific patterns.

Briefly, results show that proportions of individuals with substantial symptoms of depression range from 0.14 to 0.50 across countries, while symptoms of anxiety range from 0.13 to 0.48. Low and lower-middle-income non-Western countries tend to have higher proportions of both outcomes compared with higher-income, predominantly Western nations. Symptoms of depression and anxiety also vary across age, gender, marital status, education, employment status, religious service attendance, and immigration status in one or more countries. The results of random effects meta-analyses show that several demographic factors are significant predictors of both outcome variables when the results for all 22 countries are pooled. While being mindful of varying cultural contexts and possible translation and interpretive issues with the survey questions, the results suggest substantial variations in symptoms of both depression and anxiety across nations and key demographic groups.

## Methods

The description of the methods below has been adapted from VanderWeele, Johnson et al.[27]. Further methodological detail is available elsewhere[28,29,62–64].

### Data

Data come from the GFS, which examines the distribution and determinants of well-being across a sample of 202,898 participants from 22 geographically and culturally diverse countries. Wave 1 collected nationally-representative data from the following countries and territories: Argentina, Australia, Brazil, Egypt, Germany, Hong Kong (Special Administrative Region of China, with mainland China also included from 2024 onwards), India, Indonesia, Israel, Japan, Kenya, Mexico, Nigeria, the Philippines, Poland, South Africa, Spain, Sweden, Tanzania, Turkey, the United Kingdom, and the United States. These countries were chosen to: (a) maximize coverage of the world's population; (b) ensure geographic, cultural, and religious diversity; and (c) prioritize feasibility and existing data collection infrastructure. Gallup Inc. conducted the data collection primarily in 2023, although some regions began in 2022; timing varied by country, and more information can be found elsewhere[28]. The precise sampling design to ensure nationally-representative samples varied by country[28]. Plans are in place to collect four additional waves of annual panel data on the participants from 2024 to 2027. The translation process followed the TRAPD model (translation, review, adjudication, pretesting, and documentation) for cross-cultural research (ccsg.isr.umich.edu/chapters/translation/overview). Gallup began by translating the questionnaire for cognitive interviews and pilot testing. The translated documents were then evaluated by scholars

in participating countries to determine whether they accurately reflected original question meanings and would measure relevant constructs in the intended manner. The instruments were then tested with respondents in each GFS country and territory. Ten cognitive interviews (CIs) were completed in each country except India, where 20 were completed. Interviewers assessed how well participants understood each question, and identified issues with question wording and difficulty. Multiple versions of some questions were discussed so that comparisons in question wording and response options could be evaluated. Revised questionnaires were then pretested in each country to determine whether the planned data collection process was feasible and efficient. About 50 pretests were administered in each country except India, where 101 were conducted. Additional details are documented in the GFS Questionnaire Development Report[29], Methodology[28], Codebook (https://osf.io/cg76b), and Translations documents[62]. Data are publicly available through COS.

### Measures

**Dependent variables.** Symptoms of depression and anxiety were measured with the four-item Patient Health Questionnaire for Anxiety and Depression—PHQ-4[65]. This measure was chosen because it is brief, easy to understand, has been used in diverse populations, and is effective for monitoring and detection of potential mental health problems at the population level[66–69]. Respondents were asked: "Over the last two weeks, how often have you been bothered by the following problems: (a) feeling down, depressed, or hopeless; (b) little interest or pleasure in doing things; (c) feeling nervous, anxious, or on edge; and (d) not being able to stop or control worrying?" Response categories were 0 = not at all, 1 = several days, 2 = more than half the days, and 3 = nearly every day. A measure indicating substantial symptoms of depression was constructed by adding the scores on the first two items together, and then creating a dichotomous variable coded 1 if the combined score was greater than or equal to 3 and 0 if it was less than 3[65]. An indicator for substantial symptoms of anxiety was created in the same way using the last two items. These cut-off points are not definitive indications of depression and anxiety, but rather are the cut-offs often used to indicate the need for additional screening and assessment. Cronbach's alpha estimates for depression and anxiety were 0.74 and 0.79, respectively, for all countries combined. For each country separately, the estimates were: Argentina = 0.77, 0.79; Australia = 0.82, 0.85; Brazil = 0.75, 0.79; Egypt = 0.61, 0.74; Germany = 0.85, 0.74; Hong Kong = 0.68, 0.84; India = 0.45, 0.67; Indonesia = 0.60, 0.78; Israel = 0.75, 0.82; Japan = 0.85, 0.86; Kenya = 0.58, 0.63; Mexico = 0.76, 0.76; Nigeria = 0.62, 0.72; the Philippines = 0.54, 0.66; Poland = 0.73, 0.79; South Africa = 0.58, 0.60; Spain = 0.75, 0.78; Sweden = 0.78, 0.88; Tanzania = 0.53, 0.73; Turkey = 0.76, 0.78; the United Kingdom = 0.83, 0.87; and the United States = 0.84, 0.85.

**Demographic variables.** Continuous age was classified as: 18–24, 25–29, 30–39, 40–49, 50–59, 60–69, 70–79, and 80 or older. Gender was assessed as male, female, or other. Marital status was assessed as single/never married, married, separated, divorced, widowed, or domestic partner. Employment was assessed as employed, self-employed, retired, student, homemaker, unemployed and searching, and other. Education was assessed as up to 8 years, 9–15 years, and 16+ years. Religious service attendance was assessed as more than once/week, once/week, one to three times/month, a few times/year, or never. Immigration status was dichotomously assessed with: "Were you born in this country, or not?" Religious tradition was measured with categories of Christianity, Islam, Hinduism, Buddhism, Judaism, Sikhism, Baha'i, Jainism, Shinto, Taoism, Confucianism, Primal/Animist/Folk religion, Spiritism, African-Derived, some other religion, or no religion/atheist/agnostic; precise response categories varied by country[62]. Racial/ethnic identity was assessed in some but not all countries, with response categories varying by country.

### Analyses

Descriptive statistics for the full sample, weighted to be nationally-representative within each country, were estimated for each of the demographic variables to document variation on each measure, demonstrate the representative nature of the data, and allow comparisons with other surveys. Nationally-representative proportions of depression and anxiety were estimated separately for each country and ordered from highest to lowest, along with 95% confidence intervals and standard deviations. Variations in proportions for depression and anxiety across demographic categories were estimated, with all analyses initially conducted by country (see the Supplementary Data file). Primary results consisted of random effects meta-analyses of country-specific proportions of depression and anxiety in each specific demographic category[70,71], along with 95% confidence intervals, standard errors, upper and lower limits of a 95% prediction interval across countries, heterogeneity ($\tau$), and $I^2$ for evidence concerning variation within a particular demographic variable across countries[72]. Forest plots of estimates are available in the Supplementary Data. The meta-analyses were conducted in **R** (R Core Team, 2024) using the "metafor" package[73]. Within each country, a global test of variation of depression and anxiety across levels of each particular demographic variable was conducted, and a pooled $p$-value across countries was reported concerning evidence for variation within any country[74]. Bonferroni corrected $p$-value thresholds were provided based on the number of demographic variables[75,76]. Two-tailed tests were used. Country-specific proportions of depression and anxiety by religious tradition and race/ethnicity were estimated whenever the variables were available (see the Supplementary Data), but these variables were not included in the meta-analyses because response categories varied by country. As supplementary analyses, population-weighted meta-analyses were also conducted. All analyses were pre-registered with COS prior to data access, and code to reproduce them is openly available in an online repository[77].

### Missing data

Missing data on all variables was imputed using multivariate imputation by chained equations, and five imputed datasets were used[78,79]. To account for variation in the assessment of certain variables across countries (e.g., religious tradition and race/ethnicity), the imputation process was conducted separately in each country. This within-country imputation approach ensured that the imputation models accurately reflected country-specific contexts and assessment methods. Sampling weights were included in the imputation models to account for specific-variable missingness that may have been related to the probability of inclusion in the study.

### Accounting for complex sampling design

The GFS used different sampling designs across countries based on the availability of existing panels and recruitment needs[28]. All analyses accounted for the complex survey design components by including weights, primary sampling units, and strata. Additional methodological detail, including accounting for the complex sampling design, is provided elsewhere[64,77].

### Ethics approval and informed consent

Ethical approval was granted by the Institutional Review Boards at Baylor University (IRB Reference #: 1841317) and Gallup Inc. (IRB Reference #: 2021-11-02). Gallup is a multi-national corporation and its IRB covers all countries included in the GFS. All participants provided informed consent to Gallup and IRB approval for all data collection activities was obtained by Gallup (https://doi.org/10.1007/s10654-024-01167-9). IRB approval for data analysis was granted by Baylor University. All personally identifiable information (PII) was removed from the data used in this study by Gallup, and was not accessible to the authors. This research conformed to the principles of the Helsinki Declaration.

### Results

#### Descriptive statistics

Table 1 provides descriptive statistics for all variables for the 22 countries combined. Age ranged from 18 to 80 +, and gender was almost equally distributed among women (51%) and men (49%), with a very small number

## Table 1 | Nationally-representative descriptive statistics of the observed sample

| Variable | Proportion | Frequency |
|---|---|---|
| **Age** | | |
| 18–24 | 0.13 | 27,007 |
| 25–29 | 0.10 | 20,700 |
| 30–39 | 0.20 | 40,256 |
| 40–49 | 0.17 | 34,464 |
| 50–59 | 0.16 | 31,793 |
| 60–69 | 0.14 | 27,763 |
| 70–79 | 0.08 | 16,776 |
| 80 or older | 0.02 | 4119 |
| Missing | 0.00 | 20 |
| **Gender** | | |
| Male | 0.49 | 98,411 |
| Female | 0.51 | 103,488 |
| Other | 0.00 | 602 |
| Missing | 0.00 | 397 |
| **Marital status** | | |
| Single/never been married | 0.26 | 52,115 |
| Married | 0.53 | 107,354 |
| Separated | 0.03 | 5195 |
| Divorced | 0.06 | 11,654 |
| Widowed | 0.05 | 9823 |
| Domestic partner | 0.07 | 14,931 |
| Missing | 0.01 | 1826 |
| **Employment** | | |
| Employed by an employer | 0.39 | 78,815 |
| Self-employed | 0.18 | 36,362 |
| Retired | 0.14 | 29,303 |
| Student | 0.05 | 10,726 |
| Homemaker | 0.11 | 21,677 |
| Unemployed and looking for a Job | 0.08 | 16,790 |
| None of these/other | 0.04 | 8431 |
| Missing | 0.00 | 793 |
| **Education** | | |
| Up to 8 years | 0.22 | 45,078 |
| 9–15 years | 0.57 | 115,096 |
| 16+ years | 0.21 | 42,578 |
| Missing | 0.00 | 146 |
| **Service attendance** | | |
| >1/week | 0.13 | 26,537 |
| 1/week | 0.19 | 39,157 |
| 1–3/month | 0.10 | 19,749 |
| A few times a year | 0.20 | 41,436 |
| Never | 0.37 | 75,297 |
| Missing | 0.00 | 722 |
| **Immigration status** | | |
| Born in this country | 0.94 | 190,998 |
| Born in another country | 0.05 | 9791 |
| Missing | 0.01 | 2110 |

Notes: Country-specific descriptive statistics are available in the Supplementary Data file; Data = Global Flourishing Study, wave 1, weighted.

of other gender identities (<1%). A majority of respondents were married (53%), about 39% were employed by an employer, and roughly 57% attained 9–15 years of education. For religious service attendance, 37% never attended, 20% did a few times a year, 19% reported once a week, 13% said more than once a week, and 10% said 1–3 times a month. Most participants (94%) were native-born. Turkey had the smallest representation (1%), and the United States had the largest (19%). Nationally-representative descriptive statistics for each individual country are provided in Tables S1–S88 (odd numbered tables) in the Supplementary Data file.

### Symptoms of depression and anxiety across countries

Table 2 shows proportions of individuals with substantial symptoms of depression for all 22 countries combined and for each country separately in descending order so that readers can easily observe the cross-national variation. In the full sample, the proportion was 0.33 [95% CI: 0.32, 0.34], while individual country proportions ranged from 0.14 [0.12, 0.16] in Poland to 0.50 [0.49, 0.52] in the Philippines. Four of the five highest proportions occurred in lower-income, non-Western countries (the Philippines, India, Tanzania, and Nigeria), while four of the five lowest were in higher-income, predominantly Western nations (the United States, Sweden, Germany, and Poland). Standard deviations were lowest in Poland (0.35) and Germany (0.36), and highest in the Philippines, India, Tanzania, Hong Kong, and Nigeria (all were 0.50). Table 3 shows the findings for anxiety, which were similar to depression. In all countries combined, the proportion with more substantial anxiety symptoms was 0.30 [0.29, 0.31], and it ranged from 0.13 [0.11, 0.14] in Poland to 0.48 [0.47, 0.50] in the Philippines. The highest proportions occurred in the Philippines, Brazil, Egypt, Turkey, and Argentina, while the lowest were found in Japan, Israel, Sweden, Indonesia, and Poland. Standard deviations were lowest in Poland (0.33) and Indonesia (0.35), and highest in the Philippines, Brazil, and Egypt (all three

## Table 2 | Ordered proportions of each country (symptoms of depression)

| Country | Proportion | LCI | UCI | SD |
|---|---|---|---|---|
| Philippines | 0.50 | 0.49 | 0.52 | 0.50 |
| India | 0.49 | 0.48 | 0.50 | 0.50 |
| Tanzania | 0.45 | 0.43 | 0.48 | 0.50 |
| Hong Kong | 0.45 | 0.43 | 0.48 | 0.50 |
| Nigeria | 0.45 | 0.43 | 0.47 | 0.50 |
| Turkey | 0.43 | 0.39 | 0.46 | 0.49 |
| Brazil | 0.41 | 0.40 | 0.42 | 0.49 |
| Kenya | 0.40 | 0.38 | 0.41 | 0.49 |
| Egypt | 0.36 | 0.34 | 0.38 | 0.48 |
| Argentina | 0.36 | 0.34 | 0.37 | 0.48 |
| South Africa | 0.32 | 0.30 | 0.35 | 0.47 |
| United Kingdom | 0.29 | 0.27 | 0.31 | 0.45 |
| Spain | 0.29 | 0.28 | 0.30 | 0.45 |
| Mexico | 0.28 | 0.26 | 0.29 | 0.45 |
| Australia | 0.24 | 0.23 | 0.26 | 0.43 |
| Japan | 0.22 | 0.21 | 0.22 | 0.41 |
| Israel | 0.21 | 0.19 | 0.24 | 0.41 |
| United States | 0.20 | 0.19 | 0.21 | 0.40 |
| Indonesia | 0.19 | 0.18 | 0.20 | 0.39 |
| Sweden | 0.18 | 0.17 | 0.19 | 0.39 |
| Germany | 0.16 | 0.15 | 0.16 | 0.36 |
| Poland | 0.14 | 0.12 | 0.16 | 0.35 |

*LCI* lower 95% confidence interval, *UCI* upper 95% confidence interval, *SD* standard deviation.

**Table 3 | Ordered proportions of each country (symptoms of anxiety)**

| Country | Proportion | LCI | UCI | SD |
|---|---|---|---|---|
| Philippines | 0.48 | 0.47 | 0.50 | 0.50 |
| Brazil | 0.46 | 0.45 | 0.48 | 0.50 |
| Egypt | 0.44 | 0.42 | 0.46 | 0.50 |
| Turkey | 0.42 | 0.39 | 0.45 | 0.49 |
| Argentina | 0.41 | 0.39 | 0.42 | 0.49 |
| Nigeria | 0.37 | 0.35 | 0.39 | 0.48 |
| Kenya | 0.36 | 0.34 | 0.37 | 0.48 |
| India | 0.36 | 0.35 | 0.37 | 0.48 |
| Tanzania | 0.32 | 0.30 | 0.34 | 0.47 |
| Spain | 0.32 | 0.30 | 0.33 | 0.47 |
| South Africa | 0.29 | 0.27 | 0.32 | 0.45 |
| Mexico | 0.29 | 0.28 | 0.31 | 0.45 |
| United Kingdom | 0.29 | 0.27 | 0.31 | 0.45 |
| Hong Kong | 0.29 | 0.26 | 0.31 | 0.45 |
| United States | 0.24 | 0.23 | 0.26 | 0.43 |
| Australia | 0.23 | 0.21 | 0.25 | 0.42 |
| Germany | 0.19 | 0.18 | 0.20 | 0.39 |
| Japan | 0.19 | 0.18 | 0.19 | 0.39 |
| Israel | 0.16 | 0.13 | 0.18 | 0.36 |
| Sweden | 0.16 | 0.15 | 0.16 | 0.36 |
| Indonesia | 0.14 | 0.13 | 0.15 | 0.35 |
| Poland | 0.13 | 0.11 | 0.14 | 0.33 |

*LCI* lower 95% confidence interval, *UCI* upper 95% confidence interval, *SD* standard deviation.

were 0.50). It is important to note that 95% confidence intervals for some countries overlapped. Tables S89 and S90 in the Supplementary Data file provide results for depression and anxiety treated as continuous variables. The results were comparable, although some countries switched places.

**Demographic correlates of symptoms of depression and anxiety**
Tables 4 and 5 show results from random effects meta-analyses of country-specific proportions for all 22 countries combined for each demographic category, with each country given equal weight regardless of population size (population-weighted analyses are provided in Tables S93 and S94 in the Supplementary Data file, and discussed below). Proportions, 95% confidence intervals (CI), standard errors (SE), lower (LL) and upper limits (UL) of prediction intervals, heterogeneity ($\tau$), $I^2$, and global $p$-values were computed separately for each variable. These allow readers to assess the associations between each demographic variable and the outcome measures for all 22 countries combined and for each nation individually.

The results for depression, which were pooled across all 22 countries, are shown in Table 4. There was a progressive decline in proportions of depression as age increased, from 0.40 [0.34, 0.45] for the 18-24 age group to 0.16 [0.07, 0.31] for those 80 or older. Women had a slightly higher proportion of depression compared with men (0.32 and 0.29, respectively), but this difference was relatively small. In terms of marital status, domestic partners had the lowest proportion of depression symptoms (0.22, [0.11, 0.40]), closely followed by married (0.26, [0.21, 0.32]). The highest proportion occurred for separated (0.41, [0.36, 0.46]). When considering employment status, retired had the lowest proportion (0.24, [0.20, 0.30]), and unemployed and looking for a job had the highest (0.42, [0.37, 0.47]). Proportions of depression symptoms decreased as years of education increased, from 0.35 [0.30, 0.41] for up to 8 years to 0.26 [0.21, 0.31] for

16+ years, but these differences were relatively small. Differences in proportions of depression symptoms across levels of religious service attendance and immigration status were small. Results were similar when depression was treated as a continuous variable (see Table S91).

Table 5 shows the results for anxiety symptoms. There was a decline in proportions as age increased, from 0.38 [0.32, 0.44] to 0.09 [0.04, 0.22] for the youngest to oldest age groups. There was a small gender difference for women (0.31 [0.26, 0.36]) and men (0.26 [0.22, 0.31]). Domestic partner had the lowest proportion of anxiety symptoms (0.15, [0.06, 0.33]) among the marital status groups, and separated had the highest (0.40, [0.34, 0.45]). For employment status, retired had the lowest proportion (0.20, [0.16, 0.25]), and unemployed and looking for a job had the highest (0.39, [0.34, 0.44]). The proportion for anxiety symptoms was lower among those with 16+ years of education (0.23, [0.19, 0.27]) compared with up to 8 years (0.30, [0.24, 0.37]) and 9-15 years (0.29, [0.24, 0.33]). Religious service attendance was not notably associated with anxiety, and the differences between categories were relatively small. Immigrants had a slightly higher proportion of anxiety symptoms (0.32, [0.27, 0.36]) compared with native-born individuals (0.28, [0.24, 0.33]). When anxiety was treated as a continuous variable (see Table S92), the results were comparable.

Supplementary Data Tables S93 and S94 complement these results by providing population-weighted meta-analyses, where each country's results were weighted according to its actual 2023 population size. This means that India had a greater influence on the results because it was the largest country included in the study. Compared with Table 4, the patterns for age, gender, employment, education, religious service attendance, and immigration status were comparable for depression symptoms, although the proportions were somewhat different (Table S93). The results for marital status were slightly different. In Table 3, the lowest proportion of depression symptoms was observed for a domestic partner, but in the population-weighted findings, the lowest was married. For anxiety (Table S94), the findings were comparable to Table 5 with one exception: for marital status, widowed had the lowest proportion, not domestic partner. Married and divorced rates were also lower than domestic partners, but these differences were very small.

**Differences in demographic correlates across countries**
Tables 4 and 5 also provide information about variation in these associations across countries. The global $p$-value for each set of demographic characteristics was significant beyond the Bonferroni corrected threshold of 0.007 for each set of variables for both depression and anxiety symptoms, indicating that each was significant in at least one country. Heterogeneity ($\tau$) statistics provide an estimate of how much mental health scores in each demographic category varied across countries (larger numbers indicate more variation). When evaluating age groups, $\tau$ estimates were considerably higher for the 80 or older age group for both outcomes, indicating that there was more variation in proportions of depression and anxiety symptoms across countries in this category. Heterogeneity estimates were similar for women and men for both outcomes. The $\tau$ was much higher for domestic partners than for any other marital status group, meaning that proportions for both depression and anxiety symptoms varied more across countries among this group than they did for the other categories. Heterogeneity was similar (0.10–0.14) for all categories of employment status. For education categories, they were almost identical (0.11–0.12) for depression symptoms, but for anxiety symptoms, the $\tau$ was somewhat higher among those with lower levels of education. There was relatively little variation in $\tau$ estimates for categories of religious service attendance, but they were slightly lower among never attended. They were also similar for immigrants and native-born individuals (0.09–0.12). Supplementary Data Tables S1–S88 (even numbered tables) parallel Tables 4 and 5 but for each country separately, and provide additional insight into country-specific variations in depression and anxiety symptoms across demographic characteristics (see the Forest Plots in the Supplementary Data file as well). Further discussion and key results from these tables are provided below.

**Table 4 | Random effects meta-analysis of symptoms of depression proportions by demographic category**

| Variable | Category | Proportion | 95% CI of proportion | SE analog (CI width/4) | Prediction interval | | Heterogeneity (τ) | $I^2$ | Global p-value |
|---|---|---|---|---|---|---|---|---|---|
| | | | | | LL | UL | | | |
| Age group | | | | | | | | | <0.001** |
| | 18–24 | 0.40 | (0.34, 0.45) | 0.03 | 0.14 | 0.64 | 0.13 | 92.4 | |
| | 25–29 | 0.35 | (0.30, 0.41) | 0.03 | 0.14 | 0.56 | 0.13 | 92.9 | |
| | 30–39 | 0.33 | (0.28, 0.38) | 0.03 | 0.14 | 0.49 | 0.12 | 92.2 | |
| | 40–49 | 0.30 | (0.25, 0.35) | 0.02 | 0.13 | 0.50 | 0.11 | 91.8 | |
| | 50–59 | 0.28 | (0.23, 0.34) | 0.03 | 0.14 | 0.51 | 0.12 | 93.2 | |
| | 60–69 | 0.25 | (0.21, 0.31) | 0.03 | 0.12 | 0.49 | 0.12 | 93.6 | |
| | 70–79 | 0.23 | (0.18, 0.30) | 0.03 | 0.09 | 0.55 | 0.14 | 95.3 | |
| | 80 or older | 0.16 | (0.07, 0.31) | 0.06 | 0.00 | 0.51 | 0.28 | 99.3 | |
| Gender | | | | | | | | | <0.001** |
| | Male | 0.29 | (0.25, 0.35) | 0.03 | 0.13 | 0.48 | 0.12 | 92.3 | |
| | Female | 0.32 | (0.27, 0.37) | 0.03 | 0.16 | 0.52 | 0.12 | 92.1 | |
| | Other | 0.20 | (0.04, 0.60) | 0.14 | 0.00 | 1.00 | 0.68 | 99.8 | |
| Marital status | | | | | | | | | <0.001** |
| | Married | 0.26 | (0.21, 0.32) | 0.03 | 0.11 | 0.48 | 0.13 | 94.2 | |
| | Separated | 0.41 | (0.36, 0.46) | 0.03 | 0.21 | 0.61 | 0.12 | 90.9 | |
| | Divorced | 0.35 | (0.28, 0.42) | 0.03 | 0.15 | 0.67 | 0.16 | 95.3 | |
| | Widowed | 0.30 | (0.25, 0.36) | 0.03 | 0.14 | 0.53 | 0.12 | 93.1 | |
| | Domestic partner | 0.22 | (0.11, 0.40) | 0.07 | 0.00 | 0.67 | 0.35 | 99.3 | |
| | Single, never married | 0.36 | (0.32, 0.41) | 0.02 | 0.18 | 0.56 | 0.10 | 88.3 | |
| Employment status | | | | | | | | | <0.001** |
| | Employed by an employer | 0.30 | (0.25, 0.35) | 0.03 | 0.14 | 0.52 | 0.12 | 92.3 | |
| | Self-employed | 0.29 | (0.24, 0.35) | 0.03 | 0.13 | 0.58 | 0.13 | 93.9 | |
| | Retired | 0.24 | (0.20, 0.30) | 0.02 | 0.12 | 0.46 | 0.11 | 92.9 | |
| | Student | 0.37 | (0.32, 0.42) | 0.03 | 0.13 | 0.58 | 0.13 | 92.3 | |
| | Homemaker | 0.33 | (0.28, 0.39) | 0.03 | 0.15 | 0.51 | 0.12 | 92.2 | |
| | Unemployed and looking for a job | 0.42 | (0.37, 0.47) | 0.02 | 0.17 | 0.58 | 0.11 | 90.1 | |
| | None of these/other | 0.37 | (0.32, 0.43) | 0.03 | 0.16 | 0.60 | 0.13 | 92.9 | |
| Education | | | | | | | | | <0.001** |
| | Up to 8 years | 0.35 | (0.30, 0.41) | 0.03 | 0.19 | 0.58 | 0.12 | 91.7 | |
| | 9–15 years | 0.31 | (0.27, 0.36) | 0.02 | 0.14 | 0.49 | 0.11 | 91.0 | |
| | 16+ years | 0.26 | (0.21, 0.31) | 0.03 | 0.12 | 0.49 | 0.12 | 93.5 | |
| Religious service attendance | | | | | | | | | <0.001** |
| | >1/week | 0.32 | (0.27, 0.39) | 0.03 | 0.12 | 0.67 | 0.14 | 94.2 | |
| | 1/week | 0.34 | (0.28, 0.40) | 0.03 | 0.13 | 0.61 | 0.14 | 94.0 | |
| | 1–3/month | 0.35 | (0.29, 0.40) | 0.03 | 0.15 | 0.54 | 0.13 | 93.0 | |
| | A few times a year | 0.29 | (0.24, 0.35) | 0.03 | 0.12 | 0.51 | 0.13 | 93.7 | |
| | Never | 0.30 | (0.26, 0.34) | 0.02 | 0.16 | 0.46 | 0.09 | 88.7 | |
| Immigration status | | | | | | | | | <0.001** |
| | Born in this country | 0.31 | (0.26, 0.36) | 0.03 | 0.14 | 0.50 | 0.12 | 92.2 | |
| | Born in another country | 0.32 | (0.28, 0.36) | 0.02 | 0.20 | 0.57 | 0.09 | 86.9 | |

Notes: *p .05; **p .007 (Bonferroni corrected threshold); CI 95% confidence interval, SE Standard error, LL Lower limit, UL Upper limit, Global p-value global F (Wald) test for the overall joint significance of each set of indicator variables (two-tailed tests).

## Discussion

Relatively few studies have used large nationally-representative samples and standardized research designs to examine how symptoms of depression and anxiety vary across countries and demographic groups around the world[5–7,13,16–20,22,23]. The current study addressed this limitation by analyzing data from 22 diverse countries. In general, lower-income, non-Western countries tended to have higher proportions of both outcomes compared with higher-income, predominantly Western nations. These differences should not be overstated or interpreted as precise rankings, however, due to varying cultural contexts and possible interpretations or responses to the

**Table 5 | Random effects meta-analysis of symptoms of anxiety proportions by demographic category**

| Variable | Category | Proportion | 95% CI of proportion | SE analog (CI width/4) | Prediction interval | | Heterogeneity (τ) | I² | Global p-value |
|---|---|---|---|---|---|---|---|---|---|
| | | | | | LL | UL | | | |
| Age group | | | | | | | | | <0.001** |
| | 18–24 | 0.38 | (0.32, 0.44) | 0.03 | 0.13 | 0.62 | 0.14 | 93.4 | |
| | 25–29 | 0.34 | (0.29, 0.40) | 0.03 | 0.13 | 0.58 | 0.13 | 93.0 | |
| | 30–39 | 0.32 | (0.27, 0.37) | 0.02 | 0.14 | 0.48 | 0.11 | 91.2 | |
| | 40–49 | 0.29 | (0.24, 0.33) | 0.02 | 0.13 | 0.46 | 0.10 | 90.9 | |
| | 50–59 | 0.26 | (0.21, 0.31) | 0.02 | 0.12 | 0.48 | 0.10 | 91.9 | |
| | 60–69 | 0.22 | (0.17, 0.28) | 0.03 | 0.08 | 0.49 | 0.12 | 94.4 | |
| | 70–79 | 0.20 | (0.15, 0.27) | 0.03 | 0.05 | 0.48 | 0.14 | 96.2 | |
| | 80 or older | 0.09 | (0.04, 0.22) | 0.05 | 0.00 | 0.46 | 0.20 | 99.4 | |
| Gender | | | | | | | | | <0.001** |
| | Male | 0.26 | (0.22, 0.31) | 0.02 | 0.12 | 0.44 | 0.10 | 91.3 | |
| | Female | 0.31 | (0.26, 0.36) | 0.03 | 0.14 | 0.52 | 0.12 | 92.4 | |
| | Other | 0.13 | (0.02, 0.56) | 0.14 | 0.00 | 1.00 | 0.57 | 99.9 | |
| Marital status | | | | | | | | | <0.001** |
| | Married | 0.24 | (0.20, 0.30) | 0.03 | 0.10 | 0.48 | 0.12 | 93.7 | |
| | Separated | 0.40 | (0.34, 0.45) | 0.03 | 0.17 | 0.67 | 0.13 | 92.2 | |
| | Divorced | 0.29 | (0.24, 0.35) | 0.03 | 0.14 | 0.63 | 0.12 | 93.3 | |
| | Widowed | 0.25 | (0.19, 0.31) | 0.03 | 0.07 | 0.49 | 0.13 | 94.9 | |
| | Domestic partner | 0.15 | (0.06, 0.33) | 0.07 | 0.00 | 0.52 | 0.31 | 99.5 | |
| | Single, never married | 0.33 | (0.29, 0.38) | 0.02 | 0.16 | 0.52 | 0.10 | 89.4 | |
| Employment status | | | | | | | | | <0.001** |
| | Employed by an employer | 0.28 | (0.24, 0.33) | 0.02 | 0.13 | 0.47 | 0.10 | 91.3 | |
| | Self-employed | 0.27 | (0.22, 0.32) | 0.03 | 0.11 | 0.45 | 0.12 | 93.6 | |
| | Retired | 0.20 | (0.16, 0.25) | 0.02 | 0.07 | 0.40 | 0.11 | 94.0 | |
| | Student | 0.35 | (0.29, 0.41) | 0.03 | 0.10 | 0.57 | 0.14 | 93.7 | |
| | Homemaker | 0.31 | (0.27, 0.37) | 0.03 | 0.15 | 0.54 | 0.12 | 92.4 | |
| | Unemployed and looking for a job | 0.39 | (0.34, 0.44) | 0.02 | 0.20 | 0.59 | 0.10 | 89.0 | |
| | None of these/other | 0.34 | (0.29, 0.40) | 0.03 | 0.14 | 0.59 | 0.13 | 93.1 | |
| Education | | | | | | | | | <0.001** |
| | Up to 8 years | 0.30 | (0.24, 0.37) | 0.03 | 0.05 | 0.51 | 0.15 | 95.3 | |
| | 9–15 years | 0.29 | (0.24, 0.33) | 0.02 | 0.13 | 0.48 | 0.11 | 91.2 | |
| | 16+ years | 0.23 | (0.19, 0.27) | 0.02 | 0.12 | 0.41 | 0.09 | 89.9 | |
| Religious service attendance | | | | | | | | | <0.001** |
| | >1/week | 0.29 | (0.24, 0.35) | 0.03 | 0.11 | 0.52 | 0.13 | 93.8 | |
| | 1/week | 0.31 | (0.26, 0.36) | 0.03 | 0.13 | 0.49 | 0.12 | 92.5 | |
| | 1–3/month | 0.32 | (0.27, 0.38) | 0.03 | 0.11 | 0.51 | 0.14 | 94.0 | |
| | A few times a year | 0.27 | (0.22, 0.32) | 0.02 | 0.13 | 0.48 | 0.11 | 92.7 | |
| | Never | 0.28 | (0.23, 0.33) | 0.02 | 0.13 | 0.46 | 0.10 | 91.4 | |
| Immigration status | | | | | | | | | <0.001** |
| | Born in this country | 0.28 | (0.24,0.33) | 0.02 | 0.13 | 0.47 | 0.11 | 91.8 | |
| | Born in another country | 0.32 | (0.27, 0.36) | 0.02 | 0.17 | 0.57 | 0.10 | 89.0 | |

Notes: *p .05; **p .007 (Bonferroni corrected threshold); CI 95% confidence interval, SE Standard error, LL Lower limit, UL Upper limit, Global p-value global F (Wald) test for the overall joint significance of each set of indicator variables (two-tailed tests).

depression and anxiety symptoms questions across countries[80,81]. They do, however, suggest that lower per capita income may be a strong predictor of poor mental health across nations. In addition, many previous studies have used clinical diagnoses and cut-off points[7,21–24]. While important, there are many additional individuals who suffer from sub-clinical levels of depression and anxiety symptoms, and these can profoundly influence their lives[25]. The current results suggest that potentially problematic levels of psychological distress may be more common than some studies imply. The results also showed that proportions of both depression and anxiety symptoms varied across demographic, socioeconomic, and religious

characteristics. Global p-values indicated that both depression and anxiety symptoms were associated with each of the characteristics examined here in at least one country. As shown in Tables S1–S88 (odd numbered tables) and the Forest Plots in the Supplementary Data file, patterns often differed across countries, and these findings are discussed in detail below.

With respect to age, there were significant differences in proportions of depression symptoms in all countries, but less so in Egypt, India, Israel, Nigeria, Poland, and Turkey, suggesting that age is an important correlate in many, but not all, nations around the world. Looking across the findings for individual countries shown in the Supplementary Data file, there was a somewhat linear decrease in proportions of depression symptoms as age increased in many nations, with one nuance: there was a slight increase in depression symptoms for 80 or older compared with 70–79 in Argentina, Australia, Brazil, Germany, Japan, the Philippines, Sweden, and the United Kingdom. Four countries (Kenya, Mexico, Spain, and Tanzania) displayed either a U-shaped relationship or no clear trend. The findings for anxiety symptoms were somewhat similar except: (a) significant and non-linear associations were observed in Egypt, India, and Turkey; (b) a somewhat linear relationship was observed in Spain; and (c) relatively null results surfaced for Kenya and the Philippines. These results are consistent with previous research showing associations between age and mental health across countries[22,26]. However, the literature broadly suggests that poor mental health tends to peak in adolescence and early adulthood, decline across adulthood, and then show the lowest prevalence among older adults. While some GFS countries followed this typical pattern, others did not, and the U-shaped associations and slight increases in poor mental health toward the end of life in higher and lower income nations are novel and merit additional attention. As life expectancy continues to increase in many parts of the world, additional resources may need to be devoted to mental health issues later in life, in addition to prevention and treatment programs aimed at vulnerable adolescents and young adults. Overall, factors that may contribute to cross-cultural differences include age at first marriage, divorce (including the ability to get one), economic and labor issues (e.g., moving to find work, chronic unemployment at different ages), access to healthcare across the life course, age-specific stressors and stress, real or perceived age discrimination, and the death of loved ones, among others[82–84]. Future research should examine how each of these contributes to age-based patterns of mental health around the world.

For gender, women had especially higher proportions of depression symptoms compared with men in four countries: Argentina, Brazil, Mexico, and Sweden. Differences were also observed in Israel, the Philippines, and Spain. The results for anxiety symptoms were somewhat stronger. Women had higher proportions than men in Argentina, Brazil, Egypt, India, the Philippines, Spain, Sweden, and the United States, and other differences were observed in Germany, Mexico, and the United Kingdom. Importantly, men did not have significantly higher depression or anxiety symptoms than women in any country. Considerable previous research suggests that women may have worse mental health compared with men, and proposed explanations include gender differences in rumination, coping styles, interpersonal orientations, stressors, and physiological risk factors[9,31]. Importantly, many GFS nations did not conform to this general pattern. In the current data, countries that did and did not have expected gender differences in depression or anxiety symptoms were diverse in terms of region of the world, income, life expectancy, and majority religious tradition, so factors that contribute to gender equality/inequality in mental health may be at least partially unique to each country. This is an important finding, and future research in each nation should seek to identify distinct, country-specific factors that shape the connection between gender and mental health.

Previous research suggests that marital status may shape mental health across nations[22], and in the GFS data, it was associated with depression symptoms in every country except Egypt, Hong Kong, Nigeria, South Africa, and Turkey. Married individuals had the lowest proportions in several countries, including Argentina, Australia (tied with widowed), Brazil, Germany (tied with domestic partner), Indonesia, Israel, Poland,

Spain, Sweden, the United Kingdom, and the United States. However, domestic partners reported the lowest depressive symptoms in some countries. The highest proportion of depression symptoms was observed for single/never been married, separated, or divorced in most countries. For anxiety symptoms, there were significant differences in every country except Hong Kong, Kenya, Nigeria, the Philippines, South Africa, and Turkey. Across nations, the marital status categories with the lowest proportions of anxiety symptoms were diverse and included: (a) widowed in Argentina (tied with divorced), Australia, Germany (tied with married), Japan, Sweden, the United Kingdom, and the United States; (b) married in Brazil, Germany (tied with widowed), Indonesia, Israel, Poland, and Spain; (c) single/never been married in Egypt and Tanzania; and (d) divorced in Argentina (tied with widowed), India, and Mexico. Proportions were highest for single/never been married (in Argentina, Australia, Brazil, Spain, and Sweden), separated (in Egypt, Germany, Indonesia, Japan, Mexico, Poland, the United Kingdom, and the United States), and widowed (in India, Israel, and Tanzania). For several countries, the findings were consistent with previous research showing: (a) protective associations of marriage for mental health[19,85,86]; and (b) that being single, separated, divorced, or widowed may be risk factors for poor mental health[85]. Numerous GFS countries did not follow this general trend, however, and several findings for anxiety symptoms were not consistent with these patterns (e.g., married individuals had the lowest proportion of anxiety symptoms in only five countries, and in several nations, single/never married, separated, or widowed had the lowest). These novel findings suggest that country-specific contexts likely play important roles in shaping the connection between marital status and mental health, and future research should seek to identify and understand these factors in each nation. Possibilities include differences across countries in the economic, family, healthcare, political, and religious conditions experienced by women.

Employment status tends to be linked with mental health[20,87], and consistent with previous research, it was associated with depression symptoms in every country except Egypt, India, Kenya, Nigeria, Poland, Tanzania, and Turkey. The retired had the lowest proportion in several countries, including Australia, Brazil, Germany, Hong Kong, Indonesia, Japan, Mexico, the Philippines, Spain (tied with employed for an employer), Sweden, the United Kingdom, and the United States. The highest proportion occurred among: (a) students in Argentina, Germany, Indonesia, South Africa, and Spain; (b) unemployed and looking for a job in Australia, Brazil, Israel, Japan, Mexico, the Philippines (tied with none of these/other), Sweden, and the United States; and (c) none of these/other in Hong Kong and the United Kingdom. There were significant findings for anxiety symptoms for every country except India, Israel, Kenya, Nigeria, the Philippines, Poland, Tanzania, and Turkey. Similar to depression, retirees had the lowest proportion of anxiety symptoms in several nations, including Argentina, Australia, Brazil, Germany, Hong Kong, Indonesia, Japan, Mexico, Spain, Sweden, the United Kingdom, and the United States. The highest proportion was observed for: (a) students in Argentina, Brazil, Germany, Mexico (tied with unemployed and looking for a job), South Africa, and Spain (tied with unemployed and looking for a job); (b) unemployed and looking for a job in Indonesia, Japan, Mexico (tied with student), Spain (tied with students), Sweden, and the United States; (c) homemakers in Egypt; (d) none of these/other in Australia and the United Kingdom; and (e) self-employed in Hong Kong. Overall, these findings are intuitive and are largely consistent with previous research. They are likely linked to financial security among the employed and retired, and economic strain for groups such as students, homemakers, and the unemployed[88]. Surprisingly, however, employment status was not significant in numerous GFS countries, mostly in lower-income nations in Africa and the Middle East. This could be due to elevated levels of financial hardship despite being employed or retired, which may lead to poor mental health at levels similar to more vulnerable groups like students and the unemployed. Future research should examine this possibility in detail.

The association between education and mental health may vary across countries[5,19], and that possibility was supported by the findings reported

here. For depression symptoms, education was significant in all countries except India, Indonesia, Israel, Kenya, Nigeria, the Philippines, Poland, and Turkey. Proportions with depression symptoms were lowest among individuals with the most education in Argentina, Australia, Brazil, Egypt, Germany, Hong Kong, Japan, Mexico, Spain, Sweden, the United Kingdom, and the United States. Some associations were not linear, however, and in Australia, South Africa, and Sweden, the middle category had the highest proportion. In contrast, the middle category had the lowest proportion in Tanzania. For anxiety symptoms, the highest education category had the lowest proportion in Argentina, Brazil, Egypt, Germany, Hong Kong, India, Japan, Mexico, the Philippines, Spain, Sweden, Tanzania, and the United States. In contrast, the lowest education category had the lowest proportion in Australia and South Africa. Greater job demands and stress among occupations that require high levels of education may offer one potential explanation for these somewhat surprising findings[89]. There was no association between education and anxiety symptoms in Indonesia, Israel, Kenya, Nigeria, Poland, Turkey, and the United Kingdom. In general, these results suggest that higher levels of education are associated with lower proportions of depression and anxiety symptoms in many, but not all, nations around the world. This is somewhat surprising given the known advantages of education (e.g., access to better jobs, problem-solving skills, a higher sense of control, etc.). There are a few common characteristics among countries where education matters and does not matter for depression and anxiety symptoms, so future research will need to focus on individual contexts to explain these complex cross-cultural associations. Overall, the findings reported here are among the first for many lower-income and non-Western nations, and more work is needed in these contexts.

Considerable research, mostly in Western and Christian-majority nations, has linked multiple aspects of religious involvement, including service attendance, with mental health, primarily although not exclusively in a salutary manner[14,32]. In the GFS data, however, proportions of depression symptoms were lower among those who attended religious services more frequently in only two countries: Israel and the United States. Proportions were actually lower among infrequent attenders in Germany, Hong Kong, Sweden, Tanzania, and the United Kingdom. In two countries (Mexico and Spain), proportions of depression symptoms were slightly higher among moderate attenders compared with the more frequent and infrequent categories. Differences between attendance categories existed in Brazil, India, and Japan, but there were no clear patterns for the associations. The results were not significant for all remaining countries. For anxiety symptoms, frequent attendance was associated with a lower proportion in Australia and the United States only. In contrast, proportions were lower among infrequent attenders in Germany, Hong Kong, Poland, Spain, Sweden, and Tanzania. Five countries (Argentina, Israel, Japan, Mexico, and Nigeria) showed a non-linear pattern, where proportions of anxiety were highest among moderate attenders compared with higher and lower levels. Overall, most of these results do not fit with the pattern described in the literature using data primarily from higher income, Western, and predominantly or historically Christian nations—i.e., that participation in public religious activities may promote better mental health[14,90,91]. Interestingly, the results for the United States (a highly religious and predominantly Christian nation) were consistent with this body of work, but those for many other countries were not. These new and novel findings demonstrate the need for cross-cultural research in this area. Religious participation appears to have unique associations with mental health in different nations and cultures, possibly due to contextual factors such as economic conditions, religious history, political structures, educational systems, and media influences.

Immigration status is another known correlate of mental health in a cross-national context[33]. In the current data, however, immigration status only mattered in a few countries. Proportions of both depression and anxiety symptoms were higher among native-born individuals compared with immigrants in Australia, but lower in Sweden. In India, depression symptoms (but not anxiety symptoms) were higher among those who were native-born. There were no significant differences in the remaining countries. This pattern of largely null findings explains the weak results for the meta-analyses of this variable. It is important to note, however, that these results do not mean that immigration status is irrelevant. Considerable previous research shows that immigrant mental health is shaped by many factors, including gender, race/ethnicity, national origin, SES, family structure, social connections and isolation, language issues, and discrimination[33]. It is possible, and perhaps likely, that these and many other factors moderate the associations between immigration status and both depression and anxiety symptoms in diverse nations and cultures around the world. Future research should examine these possibilities. The differences observed between previous findings and the GFS could also be due to the fact that the current data only includes lifetime immigration experiences, which may have occurred long before mental health was assessed.

Religious tradition was not included in the meta-analyses, but it was examined in the country-specific results. There were a large number of categories, and they varied across countries, so summarizing the findings was difficult. The full results are provided in Tables S1b–S44b, but here is a brief summary. For depression symptoms, there were significant differences in proportions across religious traditions in every country except Egypt (97% Muslim). Many traditions had small sample sizes, so the comparisons below were only made among groups that represented 5% or more of the population. The largest tradition had lower depression symptoms (by 5% or more) compared with other groups in Brazil (lower than one but similar to another), Hong Kong (lower than one but higher than another), Israel, the Philippines, Sweden, the United Kingdom, and the United States; in contrast, the largest tradition had higher proportion of depression symptoms in Brazil, Indonesia, Nigeria, and South Africa (higher than one and similar to another). For anxiety symptoms, there were also significant differences in all countries except Egypt. Among groups representing 5% of the population or more, the majority religion had a lower proportion than others in Hong Kong, Indonesia, Israel, the Philippines, Sweden, and the United States, but higher in Australia, Nigeria, and South Africa (higher than one and similar to another). There is very little research on religious affiliation and mental health in the lower-income, non-Western, and non-Christian world, so the current findings present initial knowledge in this area that can serve as a foundation for future research. For additional details on the connections between mental health and the major world religions, see the *Handbook of Religion and Health*[91].

Race/ethnicity was examined in the country-specific analyses as well. There were significant differences in proportions of depression symptoms in Australia, Brazil, Hong Kong, India, Israel, Kenya, Mexico, the Philippines, South Africa, the United Kingdom, and the United States (data were not available in Germany, Japan, Spain, and Sweden). Among groups that represented at least 5% of the population, the largest racial/ethnic category had the lowest proportion of depression symptoms compared with other groups in Brazil, Israel, South Africa, Turkey, and the United Kingdom. In India, Mexico, and the United States, the largest group had lower depression than some but not all minority categories. In Hong Kong, the majority group was higher than one minority group but lower than another, while the largest group was similar to the others in Australia and the Philippines. For anxiety symptoms, significant differences were observed in Australia, Brazil, Egypt, Hong Kong, Israel, Kenya, Mexico, Nigeria, the Philippines, Poland, South Africa, Turkey, the United Kingdom, and the United States. The lowest proportion of anxiety symptoms was observed for the largest racial/ethnic group in Brazil, Israel, Kenya, Nigeria, South Africa, Turkey, and the United Kingdom. In Mexico, the Philippines, and the United States, the largest group had lower anxiety symptoms than some but not all minority groups. In Australia and Hong Kong, the largest category had a higher proportion than at least one smaller racial/ethnic group. Full results for both depression and anxiety symptoms are available in Tables S1b–S44b. Overall, cross-national research on this topic is difficult because many race/ethnicity categories vary across countries, and some nations have a lot of different

groups. Nonetheless, it is certainly possible[36,92], and should be the focus of future studies. Moving forward, scholars with expertise on race/ethnicity in each nation should examine these findings in detail, and offer insights based on their knowledge of local cultures and contexts. Currently, we know a lot about race/ethnicity in higher-income countries like the United States and many European nations, where minority racial groups often report worse mental health, which is consistent with some of the current findings. Data is sparse in Latin America, Asia, Africa, and the Middle East, however. In some countries and cultures, race/ethnicity and mental health are sensitive topics that are challenging to study, whereas in others, there are so many racial/ethnic groups that classifying and studying them is difficult. The current GFS results are, to our knowledge, the only published findings on this topic for numerous countries, and addressing this limitation should be a priority for future research.

Future work should build on these findings in several ways. First, scholars conducting independent research in each country should compare their findings with those reported here in hopes of better understanding the distribution of mental health in each country. Second, subsequent research should also attempt to determine why both depression and anxiety symptoms are relatively high in some countries, but low in others. In addition to identifying stressful conditions that also vary across countries, research on "…emotional contagion and symptom transmission in psychopathology, i.e., the complex ways in which one person's psychological distress may yield symptoms among others in his/her close environment (p.1)[93]," may provide fresh insight by shifting the focus away from individuals and toward contexts and environments that may vary across nations and cultures. Third, future research with at least two waves of GFS data should examine how demographic factors shape longitudinal trends in depression and anxiety symptoms. Fourth, we need more research on lower-income countries and nations that are not predominantly or historically Christian. This study is among the first to report findings from large nationally-representative samples in several lower-income, non-Western, and non-Christian countries. Fifth, mounting research has linked social media use with poor mental health[94,95], but very little research has been conducted outside of higher-income nations, so future research should address this weakness in the literature. And sixth, future research should examine specific contextual factors unique to each country and culture that may account for, or moderate, observed differences in mental health across countries. These likely include: (a) variation in national income, wealth, and economic development across countries, as well as levels of economic inequality within nations; (b) differences in education; (c) fertility rates and the age structures of different populations around the world; (d) cross-cultural variation in stigma surrounding mental illnesses that may arise from religious or other sources; (e) national differences in treatment and prevention of mental health problems; (f) social welfare programs and social safety nets across countries; and (g) political systems and stability that likely shape all aspects of life including mental health[5,22,30,96,97].

This study has several strengths. First, all of the survey items were carefully chosen and evaluated by leading scholars from around the world, and then extensively pretested by Gallup personnel on the ground in each country[28,29]. Second, the GFS is a very large survey, with 202,898 participants in 22 diverse countries. Given the large size and representative nature of the samples, findings based on GFS data should offer reasonable estimates of many key constructs, including depression and anxiety symptoms, compared with research based on small, non-random, or specialized samples. Third, the sample includes nations that span the income range from low to high, and are culturally, religiously, geographically, and politically diverse. This means that the findings are relevant to many individuals and groups of people around the world.

Despite these strengths, this study also has limitations. First, it is cross-sectional and only analyzes the first wave of GFS data. The baseline survey data were released on 2/13/24, and data collection for the second wave is ongoing. The findings reported here are descriptive in nature and should not be interpreted as causal, and may not generalize beyond the specific countries examined here. Second, cross-cultural research is difficult for many reasons, including language barriers, differing norms regarding sensitive issues like health, and survey question translation and interpretation issues[81,98,99], and these were present during the GFS data collection process. As described in the study documentation[28,29,62,63], Gallup conducted extensive pretesting and translation work in hopes of obtaining comparable meanings of survey items across countries, but this was difficult because some words and concepts may not have clear analogs in different languages and countries. Third, in addition to matters of translation, additional caution is needed in interpreting cross-national differences, as these may also be influenced by different modes of assessment, differing interpretations of response scales, and seasonal effects arising from data being collected in different countries at different times of the year. Therefore, strict and direct comparisons of statistics in one country vs another should be made with caution. This is especially true for subjective assessments like depression and anxiety symptoms. The diversity of the GFS is one of its strengths, but this same diversity also highlights both the challenge, as well as the need to provide fair and accurate interpretations of findings when utilizing such a diverse sample. Fourth, the current findings are based on a four-item self-report measure of symptoms of depression and anxiety[65]. There are many alternative approaches to measuring mental health, including different questionnaires and rating scales, interviews by medical professionals, and behavioral observations and assessments[100], and each may capture unique aspects of psychological distress and well-being. Additional cross-national studies using other measures of depression or anxiety symptoms may help us understand these complex outcomes.

## Conclusion

To conclude, depression and anxiety symptoms both appear to vary across countries and demographic groups. Significant nuance and variation exist in terms of which sociodemographic measures predict these outcomes, however. This descriptive work lays the foundation for future studies on the correlates of these aspects of mental health in a global context. At the time of manuscript drafting, data collection for the second wave of panel data was well underway, which will allow scholars to begin examining the causal factors underlying variations in mental health. The results of current and future studies using GFS data will help to further shape the conversation around human flourishing, which has the potential to benefit individuals, communities, and nations around the globe.

## Data availability

This study analyzed cross-sectional data from Wave 1 of the GFS, which is publicly available through the COS (https://doi.org/10.17605/OSF.IO/3JTZ8), the host for all data collected for the study. For additional information about data access and use, see the COS website (https://www.cos.io/gfs-access-data) or contact them by email (globalflourishing@cos.io). All relevant data are available from the corresponding author.

## Code availability

All code to reproduce the analyses are openly available in an online repository hosted by the COS (https://doi.org/10.17605/osf.io/vbype). Versions are available for R, SAS, Stata, and SPSS.

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

## Acknowledgements

The authors would like to acknowledge and thank Noah Padgett, Ying Chen, Sung Joon Jang, and Koichiro Shiba for their help with the data analysis. The GFS was generously funded by the John Templeton Foundation (#61665), the Templeton Religion Trust (#1308), the Templeton World Charity Foundation (#0605), Well-Being for Planet Earth, Fetzer Institute (#4354), Well Being Trust, Paul L. Foster Family Foundation, and the David & Carol Myers Foundation. The opinions expressed in this publication are those of the authors and do not necessarily reflect the views of these organizations. The funding source had no impact on the study design; on the collection, analysis, and interpretation of data; on the writing of the manuscript; or on the decision to submit the article for publication.

## Author contributions

M.B.: Conducted the data analysis, contributed to the interpretation of the data, and drafted the original manuscript. K.S.: Contributed to reviewing and editing the manuscript. S.J.J.: Contributed to the development of code for data analysis and revision of the manuscript. B.V.K.: Review and editing. R.B.: Critical review and editing. B.R.J.: Obtained funding for the project as the Principal Investigator, led and contributed to every phase of the project, contributed to the interpretation of the data, and contributed to writing and editing the manuscript. T.J.V.: Obtained funding for the project as the Principal Investigator, led and contributed to every phase of the project, contributed to the interpretation of the data, and contributed to writing and editing the manuscript.

## Competing interests
