## [Transparent Peer Review file · Communications Medicine]

Demographic Variation in Symptoms of Depression and Anxiety Across 22 Global Flourishing Study Countries

Corresponding Author: Professor Matt Bradshaw

Version 0:

Reviewer comments:

Reviewer #1

(Remarks to the Author)

1. First, I want to compliment the authors of the study. The study provides a comprehensive discussion on Depression and Anxiety across diverse nations in 22 countries. The inclusion of both developed and non-developed countries, as well as non-Western countries with culturally different religions, provides valuable information and perspective on depression and anxiety globally.

2. The introduction was written comprehensively, providing the background of the study and clearly stating its objectives, which assists readers in understanding the study conducted.

3. Methodology.

a. Data

A larger sample size is one of the strengths of this study (202,898). Including information where the data can be assessed is appreciated since readers can look at the data if they want to. However, I think it is more appropriate if the authors briefly explain the translation process rather than asking the readers to refer to any documents. The sentence (141-143) Further details are documented in the GFS Questionnaire Development Report³², Methodology³¹ Codebook (<https://osf.io/cg76b>), and Translation documents⁶⁵, which can be taken out from the text if the translation process explains.

b. Using The Patient Health Questionnaire for Anxiety and Depression – PHQ-4 measure is appropriate for this study. However, I think it is better if the author(s) can report the reliability of the PHQ-4 in each nation since the measure was translated into different languages.

c. Data Analyses – Analyses of the data were explained in detail. Using Metafor (Meta-analysis Package for R) is appropriate for this study. However, I am wondering if there are any other statistical uses for example in reporting Descriptive statistics. If so, I think it is need to include in the explanation.

4. Result was presented clearly and answering the research objectives.

5. Discussion – The discussion part was insightful and an eye-opener for readers to understand the findings of the current study. Limitation and future research was addressed. I like how the authors discussed the importance of looking more at religious perspectives since not many studies look into it. However, I would also love it if the authors could discuss what might factors contribute to the differences in the findings.

6. Overall, the article provides new insightful and novel knowledge on anxiety and depression across nations. Including non-western and non-developing countries in this study provides knowledge on global patterns of Depression and Anxiety. I couldn't wait for the article to be published.

Reviewer #2

(Remarks to the Author)

The article is interesting and provides relevant data on mental health for the scientific community. In general, I think it is a good text and that it uses appropriate methods for the proposed research. My main concern lies in the way information is presented and organized. Some aspects to improve:

- Despite providing a good justification, reading the introduction can be confusing due to the clutter of ideas. In the middle of the text, the objectives of the study are stated, although at the end the objectives of the GFS are raised again. I suggest structuring the introduction into paragraphs by ideas so that the reader can continue to understand the content better
- Given that only the PHQ-4 is used as a dependent variable, a reasoned justification could be provided as to why this measure is used and not others and its psychometric properties
- Sometimes reference is made to certain limitations or discussion of the results in the results section, such as, "Also, as discussed further below, language translation and culture-specific issues mean that question wording may have had different connotations across countries despite a concerted effort by Gallup to minimize such effects". Although it is appreciated that the authors point out these issues, they should place them in their corresponding section to facilitate reading and not duplicate the information.
- In general, it gives the feeling that many ideas and data are repeated. For example, at the beginning of the discussion, a long paragraph is dedicated to repeating the data of the results.
- In the discussion, it is necessary to compare the data in more depth with previous data and highlight if there is any novelty or relevant contribution to what was already known. In addition, one could reflect on the sociodemographic reasons that provide differences between countries. In the current state of the discussion, it seems more like an extended description of the results. Some headings could also be considered to differentiate the main sociodemographic variables.
- I would advise adding a separate limitations section and transferring to it all the limitations highlighted throughout the text (for example, in the discussion reference is made to the use of PHQ-4 as a possible limitation). Having them all collected in one section would simplify other sections and give a cleaner view of the text.

Version 1:

Reviewer comments:

Reviewer #1

(Remarks to the Author)

Dear Authors,

Feedback for the revised manuscript version.

3. Methodology.

a. Data Translation

- The translation process followed the TRAP model explained in the revised version.

b. Using the Patient Health Questionnaire for Anxiety and Depression (PHQ-4 measure is appropriate for this study. However, I think it is better if the author(s) can report the reliability of the PHQ-4 in each nation.

- The Cronbach's alpha for each country included in the revised version.

c. Data Analyses – Analyses of the data were explained in detail. Using Metafor (Meta-analysis Package for R) is appropriate for this study. However, I am wondering if there are any other statistical uses, for example, in reporting Descriptive statistics. If so, I think it needs to be included in the explanation.

- Explanation accepted

5. Discussion – The discussion part was insightful and an eye-opener for readers to understand the findings of the current study. Limitation and future research was addressed. I like how the authors discussed the importance of looking more at religious perspectives since not many studies look into it. However, I would also love it if the authors could discuss what other factors might contribute to the differences in the findings.

- I think the information added by the authors enriched the discussion.

Congratulations.

Reviewer #2

(Remarks to the Author)

Thank you for addressing all the points I suggested in the previous review. I think the article has been significantly improved.

Reviewers' comments:

Reviewer #1 (Remarks to the Author):

1. First, I want to compliment the authors of the study. The study provides a comprehensive discussion on Depression and Anxiety across diverse nations in 22 countries. The inclusion of both developed and non-developed countries, as well as non-Western countries with culturally different religions, provides valuable information and perspective on depression and anxiety globally.

- *Thank you for the positive feedback. We purposefully designed the study to address these limitations in the literature.*

2. The introduction was written comprehensively, providing the background of the study and clearly stating its objectives, which assists readers in understanding the study conducted.

- *Thank you.*

3. Methodology.

a. Data

A larger sample size is one of the strengths of this study (202,898). Including information where the data can be assessed is appreciated since readers can look at the data if they want to.

However, I think it is more appropriate if the authors briefly explain the translation process rather than asking the readers to refer to any documents. The sentence (141-143) Further details are documented in the GFS Questionnaire Development Report³², Methodology³¹ Codebook (<https://osf.io/cg76b>), and Translation documents⁶⁵, which can be taken out from the text if the translation process explains.

- *Thank you for this suggestion. This is indeed important information to include in the text. We have revised this portion of the paper in the following way: “The translation process followed the TRAPD model (translation, review, adjudication, pretesting, and documentation) for cross-cultural survey research (ccsg.isr.umich.edu/chapters/translation/overview). Gallup began by translating the questionnaire for cognitive interviews and pilot testing. The translated documents were then evaluated by scholars in participating countries to determine whether they accurately reflected original question meanings and would measure relevant constructs in the intended manner. The instruments were then tested with respondents in each GFS country and territory. Ten cognitive interviews (CIs) were completed in each country except India, where 20 were completed. Interviewers assessed how well participants understood each question, and identified issues with question wording and difficulty. Multiple versions of some questions were discussed so that comparisons in question wording and response options could be evaluated. Revised questionnaires were then pretested in each country to determine whether the planned data collection process was feasible and efficient. About 50 pretests were administered in each country except India, where 101 were conducted.”*

b. Using The Patient Health Questionnaire for Anxiety and Depression – PHQ-4 measure is appropriate for this study. However, I think it is better if the author(s) can report the reliability of the PHQ-4 in each nation since the measure was translated into different languages.

- *This is a good idea. We have added the following text to the description of the measure: “Cronbach’s alpha estimates for depression and anxiety were 0.74 and 0.79, respectively, for all countries combined. For each country separately, the estimates were: Argentina=0.77, 0.79; Australia=0.82, 0.85; Brazil=0.75, 0.79; Egypt=0.61, 0.74; Germany=0.85, 0.74; Hong Kong=0.68, 0.84; India=0.45, 0.67; Indonesia=0.60, 0.78; Israel=0.75, 0.82; Japan=0.85, 0.86; Kenya=0.58, 0.63; Mexico=0.76, 0.76; Nigeria=0.62, 0.72; the Philippines=0.54, 0.66; Poland=0.73, 0.79; South Africa=0.58, 0.60; Spain=0.75, 0.78; Sweden=0.78, 0.88; Tanzania=0.53, 0.73; Turkey=0.76, 0.78; the United Kingdom=0.83, 0.87; and the United States=0.84, 0.85.”*

c. Data Analyses – Analyses of the data were explained in detail. Using Metafor (Meta-analysis Package for R) is appropriate for this study. However, I am wondering if there are any other statistical uses for example in reporting Descriptive statistics. If so, I think it is need to include in the explanation.

- *We have added an explanation for the descriptive statistics in the Analyses section of the text. If this is not what you had in mind, please let us know and we will be happy to make additional revisions.*

4. Result was presented clearly and answering the research objectives.

- *Thank you. We worked hard to summarize the extensive results in a clear and effective manner.*

5. Discussion – The discussion part was insightful and an eye-opener for readers to understand the findings of the current study. Limitation and future research was addressed. I like how the authors discussed the importance of looking more at religious perspectives since not many studies look into it. However, I would also love it if the authors could discuss what might factors contribute to the differences in the findings.

- *This is a great idea, and have added the following additional information to the Discussion section: “And sixth, future research should examine specific contextual factors unique to each country and culture that may account for, or moderate, observed differences in mental health across countries. These likely include: (a) variation in national income, wealth, and economic development across countries, as well as levels of economic inequality within nations; (b) differences in education; (c) fertility rates and the age structures of different populations around the world; (d) cross-cultural variation in stigma surrounding mental illnesses that may arise from religious or other sources; (e) national differences in treatment and prevention of mental health problems; (f) social welfare programs and social safety nets across countries; and (g) political systems and stability that likely shape all aspects of life including mental health.”*

6. Overall, the article provides new insightful and novel knowledge on anxiety and depression across nations. Including non-western and non-developing countries in this study provides knowledge on global patterns of Depression and Anxiety. I couldn't wait for the article to be published.

- *Thank you for the encouraging words! We appreciate your comments, which have improved the manuscript.*

Reviewer #2 (Remarks to the Author):

The article is interesting and provides relevant data on mental health for the scientific community. In general, I think it is a good text and that it uses appropriate methods for the proposed research. My main concern lies in the way information is presented and organized. Some aspects to improve:

- *Thank you for reviewing our manuscript, and for the helpful comments.*

- Despite providing a good justification, reading the introduction can be confusing due to the clutter of ideas. In the middle of the text, the objectives of the study are stated, although at the end the objectives of the GFS are raised again. I suggest structuring the introduction into paragraphs by ideas so that the reader can continue to understand the content better

- *This is a good point. We have revised the entire front end accordingly in attempt to make it less confusing and repetitive. Please let us know if we addressed your concerns in the revised version, and if you have any additional comments or suggestions.*

- Given that only the PHQ-4 is used as a dependent variable, a reasoned justification could be provided as to why this measure is used and not others and its psychometric properties

- *Thank you for pointing this out. We have added the following text to the paper: "This measure was chosen because it is brief, easy to understand, has been used in diverse populations, and is effective for monitoring and detection of potential mental health problems at the population level."*

- Sometimes reference is made to certain limitations or discussion of the results in the results section, such as, "Also, as discussed further below, language translation and culture-specific issues mean that question wording may have had different connotations across countries despite a concerted effort by Gallup to minimize such effects". Although it is appreciated that the authors point out these issues, they should place them in their corresponding section to facilitate reading and not duplicate the information.

- *We agree, and have removed this from the results and now simply include it in the Discussion.*

- In general, it gives the feeling that many ideas and data are repeated. For example, at the beginning of the discussion, a long paragraph is dedicated to repeating the data of the results.

- *Good point. We have revised the Discussion section to be less repetitive and have focused on findings that were not summarized in the Results section.*

- In the discussion, it is necessary to compare the data in more depth with previous data and highlight if there is any novelty or relevant contribution to what was already known. In addition, one could reflect on the sociodemographic reasons that provide differences between countries. In the current state of the discussion, it seems more like an extended description of the results. Some headings could also be considered to differentiate the main sociodemographic variables.

- *We appreciate these suggestions. We have added additional discussion of the results compared with previous studies. We have attempted to point out where our findings align with previous studies, where they depart, and what is novel about our data and results. We have also added subheading for the demographic variables to the Discussion section. Thank you for these helpful comments.*

- I would advise adding a separate limitations section and transferring to it all the limitations highlighted throughout the text (for example, in the discussion reference is made to the use of PHQ-4 as a possible limitation). Having them all collected in one section would simplify other sections and give a cleaner view of the text.

- *This is a great point. We have now added a separate section and moved the PHQ-4 discussion as you suggested. We included the strengths in the same section as well. If you prefer to have a separate section for each, we will be happy to make this revision.*

Reviewers' comments:

Reviewer #1 (Remarks to the Author):

Dear Authors,

Feedback for the revised manuscript version.

3. Methodology.

a. Data Translation

- The translation process followed the TRAP model explained in the revised version.

*Author Response: Thank you, we are glad the revised version is acceptable.

b. Using the Patient Health Questionnaire for Anxiety and Depression (PHQ-4 measure is appropriate for this study. However, I think it is better if the author(s) can report the reliability of the PHQ-4 in each nation.

- The Cronbach's alpha for each country included in the revised version.

*Author Response: Thank you, we are glad the revised version is acceptable.

c. Data Analyses – Analyses of the data were explained in detail. Using Metafor (Meta-analysis Package for R) is appropriate for this study. However, I am wondering if there are any other statistical uses, for example, in reporting Descriptive statistics. If so, I think it needs to be included in the explanation.

- Explanation accepted

*Author Response: Thank you, we are glad the revised version is acceptable.

5. Discussion – The discussion part was insightful and an eye-opener for readers to understand the findings of the current study. Limitation and future research was addressed. I like how the authors discussed the importance of looking more at religious perspectives since not many studies look into it. However, I would also love it if the authors could discuss what other factors might contribute to the differences in the findings.

- I think the information added by the authors enriched the discussion.

*Author Response: Thank you, we are glad the revised version is acceptable.

Congratulations.

*Author Response: Thank you!

Reviewer #2 (Remarks to the Author):

Thank you for addressing all the points I suggested in the previous review. I think the article has been significantly improved.

*Author Response: Thank you, we are glad the revised version is acceptable.